# Simulating Tree Root Water Uptake in the Frame of Sustainable Agriculture for Extreme Hyper-Arid Environments Using Modeling and Geophysical Techniques

Arya Pradipta [1,*], Nektarios N. Kourgialas [2,*], Yassir Mubarak Hussein Mustafa [3], Panagiotis Kirmizakis [1] and Pantelis Soupios [1]

1   Department of Geosciences, College of Petroleum Engineering and Geosciences, King Fahd University of Petroleum and Minerals, Dhahran 31261, Saudi Arabia; p.kirmizakis@kfupm.edu.sa (P.K.); panteleimon.soupios@kfupm.edu.sa (P.S.)
2   Water Resources, Irrigation and Environmental Geoinformatics Laboratory, Institute for Olive Tree, Subtropical Crops and Viticulture, Directorate General of Agricultural Research, Hellenic Agricultural Organization "DIMITRA", 73100 Chania, Greece
3   Interdisciplinary Research Center for Construction and Building Materials, King Fahd University of Petroleum and Minerals, Dhahran 31261, Saudi Arabia; yassir.mustafa2@kfupm.edu.sa
*   Correspondence: g201506210@kfupm.edu.sa (A.P.); kourgialas@elgo.gr (N.N.K.)

**Abstract:** In order to ensure sustainability in the agricultural sector and to meet global food needs, a particularly important challenge for our time is to investigate the possibility of increasing agricultural production in areas with extreme hyper-arid environments. Warming air temperatures and sandy soils significantly influence tree root water uptake (RWU) dynamics, making accurate estimation of RWU depth distribution and magnitude crucial for effective resource management, particularly in the context of precision irrigation within agroecosystems. This study employed two non-invasive techniques, namely HYDRUS 1D and electrical resistivity tomography (ERT), to simulate RWU under controlled experimental conditions and under an extreme hyper-arid environment. The results revealed that the highest RWU rates occurred during the morning (08:00–11:00). RWU activity predominantly concentrated in the upper soil profile (0–30 cm), and the soil water content in this area was notably lower compared to the deeper soil layers. With increasing temperature, there was a tendency for the RWU zone to shift to lower depths within the soil profile. The findings of this study could have important implications for farmers, providing valuable insights to implement irrigation water management strategies.

**Keywords:** root water uptake; climate change; agricultural geophysics; modeling; HYDRUS

## 1. Introduction

The earth critical zone can be defined as the near-surface layer of earth, ranging from the top of the vegetation canopy to the topmost groundwater zone, involving vegetation, soil, weathered bedrock, and water that are essential for supporting all types of life [1]. The range of the earth critical zone varies greatly in space, depending on tectonic history, lithology, vegetation types, and climate [2]. This zone is highly active and dynamic, where many critical natural processes occur to support the living ecosystem. Special attention should be devoted to the soil–plant–atmosphere interface that plays an essential role in mass and energy exchange in the critical zone. The interaction in this interface is regarded as complex processes that are time-, scale-, and species-dependent and spatially heterogeneous. The foundation of the soil–plant–atmosphere interface lies in the movement of water from soil into the surrounding atmosphere through a particular process called root water uptake (RWU) due to the water potential differences [3]. Optimization of RWU is important since insufficient or excessive water can lead to plant mortality.

Under warming temperatures, the hydrological states and fluxes might be altered, thus influencing crop RWU [4]. The soil water content is expected to decrease, and drought intensity would be amplified through higher evapotranspiration. Subsequently, RWU activity might go deeper to reach moister soil areas in order to resist the increased temperature [5,6]. However, other studies suggested that warming temperatures could restrict RWU from deeper moisture by lowering root activity [7]. Nevertheless, plants can also change the RWU zone at a smaller time scale without altering its root system [8]. Despite these discrepancies, it can be concluded that the changing temperature profoundly impacts RWU activity. Thus, knowledge of the depth distribution and magnitude of RWU as a response to warming temperature is beneficial for a practical recommendation of agricultural water strategies. However, monitoring RWU in practice is not an easy matter since this variable is the most difficult to observe.

The water uptake by plant roots through the soil moisture is a primary mechanism that regulates the water equilibrium in field profiles and is crucial for managing agroecosystems. Mapping the root zone soil moisture is essential due to its pivotal role in diverse environmental processes and hydrological cycles. Mapping soil moisture in the root zone enables a more comprehensive understanding of plant behavior, energy equilibrium, and water processes. It aids in observing the changes in soil moisture levels across various land-cover types, which can differ significantly in terms of their spatial and temporal variations. Moreover, mapping soil moisture at the root zone level offers valuable environmental modeling and monitoring insights, particularly in small catchment areas and regions with diverse land cover. Additionally, it allows for anticipating how ecosystems will react to variations in soil water availability, a crucial factor in determining vegetation vitality and overall ecosystem well-being.

Saudi Arabia possesses a highly varied natural terrain, making it one of the most diverse landscapes globally. Spanning over 2 million square kilometers, the Kingdom boasts a diverse array of crops including palm trees, fruit, olives, coffee beans, rice, lentils, and more. The Ministry of Environment, Water, and Agriculture in Saudi Arabia has commenced the initial stage of two projects aimed at planting 49 million fruit and lemon trees across different regions. These initiatives are part of the larger Saudi Green Initiative. By 2030 (https://www.vision2030.gov.sa/, accessed on 20 February 2024), there will be a total of 45 million fruit trees in agricultural terraces and 4 million lemon trees irrigated with renewable water. These initiatives are anticipated to account for over 50% of the country's fruit imports, in order to fulfill the objectives of the Kingdom's Vision 2030. These government programs are anticipated to make a significant contribution to the sustainable cultivation and production of fruit crops, with the goal of achieving food security and sustainable development. In addition, they will also involve the utilization of rainwater and renewable water sources for irrigation purposes, as well as replenishing groundwater reserves.

Orchards that grow tree crops are crucial for the global economy and the environment because they can remain productive for many years without requiring replanting. In addition, they possess superior adaptability to harsh climatic conditions in comparison to other crops. Nevertheless, the advent of climate change poses additional obstacles as it endangers both the cultivation of trees and the availability of water. Drip irrigation, whether on the surface or subsurface, is an irrigation technique that can conserve water and nutrients by delivering water directly to the root zone and reducing evaporation. Various irrigation designs and strategies have been experimented with to optimize the efficiency of drip irrigation under specific soil, crop, and climate conditions [9].

Conducting field tests and measurements to determine water movement requires a significant amount of financial resources and time. Consequently, conceptual and mathematical models have been developed to predict the movement of solutes and water in the vadose zone. Although field measurements are still considered valuable, uncertainties are always taken into account in the modeling technique. Thus, policymakers with an

interest in sustainable agriculture can achieve more reliable results by utilizing the most suitable model.

In order to achieve the most cost-effective and efficient solution, researchers have utilized advanced numerical models like HYDRUS 1D/2D/3D to determine the ideal combination of irrigation management and design [10]. HYDRUS is a reliable and widely used software program designed for simulating the movement of solutes, heat, and water in variably saturated porous surfaces in all dimensions. It is also particularly well-suited for sandy soils [11] such as the soil type in Saudi Arabia. Furthermore, when compared to other models of the unsaturated zone, HYDRUS offers users a greater range of choices for assessing investigations on root water uptake [12,13]. In agricultural applications, HYDRUS 1D version 4.1, as an open access software instead of HYDRUS 2D/3D which is a commercial software package, has been widely used to simulate RWU under different crop, soil, and environmental conditions and provide satisfactory results [14–16]. This numerical model has been widely applied to simulate RWU at a daily time scale. However, dynamic change simulation in the diurnal variation in RWU has rarely been reported. Understanding the diurnal variation in the RWU pattern can provide a theoretical basis for irrigation application. In addition, the parameter of direct field measurements required in HYDRUS, for instance, soil water content, is typically collected through a classical method, which is expensive, labor-intensive, destructive in nature, and provides limited spatial coverage. In contrast, non-invasive geophysical techniques such as electrical resistivity tomography (ERT) can offer explicit information on soil electrical resistivity, which can be inverted and then transformed to measure soil water content and/or soil moisture [17,18]. This is actually the novelty of the suggested research: the use of a high-resolution 3D resistivity model of the subsurface, to estimate the soil moisture without the use of a few sensors spatially distributed within the study area.

Considering the aforementioned explanations, this study explored the potential of numerical simulations and geophysical techniques to examine RWU patterns in an arid environment. In particular, we aimed at (1) identifying the diurnal variations and seasonal pattern of RWU during the drying interval between successive irrigation and (2) simulating the magnitude of RWU at different depths. Based on the above, the originality of this work lies in the fact that in extreme hyper-arid environments characterized by warming air temperatures as well as very high concentrations of sand in the soils, the use of geophysical techniques could give us accurate information on the dynamics of moisture soil versus depth (overcoming the use of soil moisture sensors, which in sandy soils are extremely difficult to operate properly as described in the Materials and Methods section below). This information in turn can be used to calibrate the soil-hydraulic parameters of the HYDRUS 1D model and simulate RWU. Estimating RWU in tree crops is particularly important information for water resource management in such hyper-arid environments.

## 2. Materials and Methods

### 2.1. Site Description and Experimental Design

The experimental site is situated in Dhahran city, the eastern coastal region of Saudi Arabia, which is characterized by a desert arid climate based on the Köppen–Geiger climate classification [19]. The mean annual temperature is 26.4 °C, while the average yearly precipitation is 84 mm. July is the hottest month in this area, with an average high of 42 °C and a low of 29 °C. Dhahran has extreme seasonal variation in terms of perceived humidity, where the muggier period occurs from May to November. During the study period, no precipitation event was recorded.

The experiment was carried out in a woody box ($192 \times 192 \times 153$ cm$^3$) filled with medium-sized sand. Soil samples were collected at five different depths and from four different locations near the tree, and grain size analysis was performed (Figure 1). The grain size analysis shows that the soil comprises sand (99.63%) and clay/silt (0.37%). This is a typical soil type for Saudi Arabia, especially in the broader area of East Province and Riyadh. Based on Liu et al. [20], the soil can be considered sandy soil with 5–12% soil

moisture. It has to be mentioned that to increase the fertility of this highly sandy soil, a 5% *v/v* compost as a soil amendment was added and enough time was applied prior the experiment to achieve proper maturation of the mixed soil within the tank.

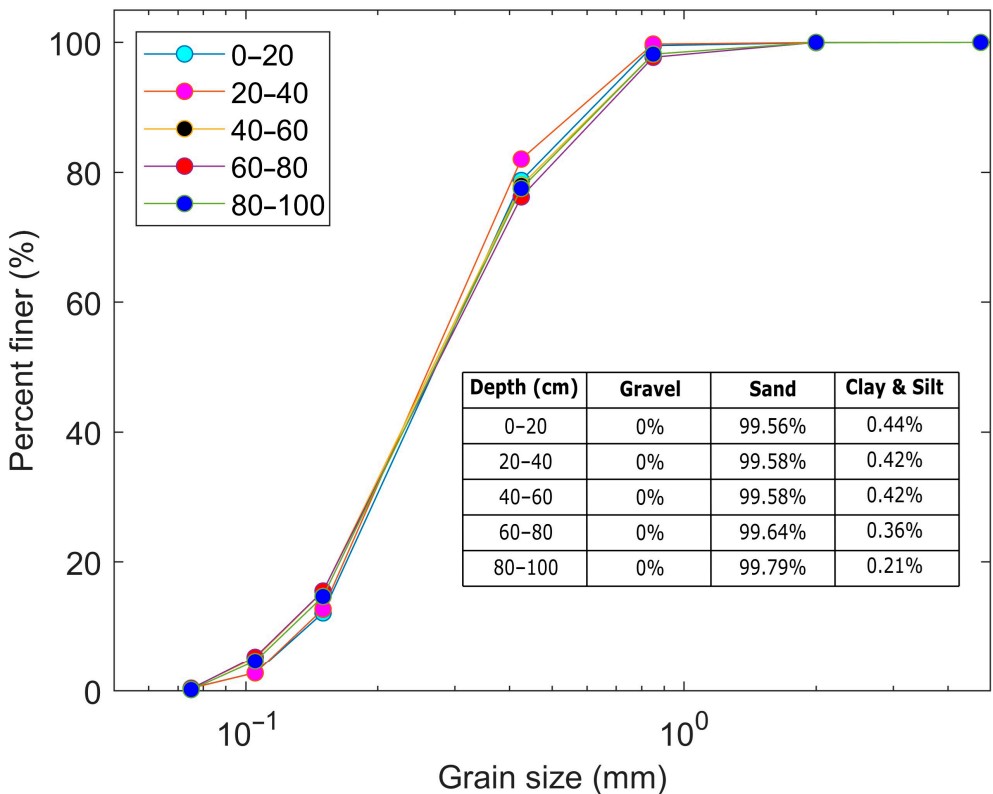

**Figure 1.** The grain size measurement and soil composition of the site location indicating homogeneity within the soil. The measurement was conducted using sieve analysis or a gradation test.

Based on the green initiative in Saudi Arabia and its priorities, the use of a citrus plant to model the RWU was decided. Citrus is widely acknowledged as a crucial horticultural crop in the global agricultural sector, with production spanning across 140 countries [21]. Citrus trees yield various fruits, including significant agricultural commodities like oranges, lemons, grapefruits, pomelos, and limes. Citrus cultivation covers a total area of 10,072,197.00 hectares worldwide and yielded a production of 158,490,986.00 tons in 2020 [22]. The majority of citrus-growing regions experience arid and sweltering summer conditions. Effective management of citrus irrigation and nutrient levels is crucial, particularly in regions where water resources are limited.

For our experimental work, a 1-year-old and 2.5 m-high grown citrus (lemon) tree was planted and monitored for one year. Citrus plants possess a superficial root system. Directing irrigation towards the efficient root zone is crucial, minimizing the amount of water that drains beyond it. Most lemon tree roots primarily reside within the uppermost 60 cm of soil, but the optimal root zone typically lies within the uppermost 30 to 40 cm of soil, with variations depending on the specific soil composition [23–25]. Thus, the target area for monitoring the root zone and characterizing its soil moisture is limited to the upper 50 cm, with the highest density of the root zone expected at a depth of 30–40 cm [26].

Drip irrigation was used for the watering of the plant during this experimental study since this method is especially favored for vegetable and perennial crops, such as fruit trees. Drip irrigation allows for precise application of water and chemicals, ensuring optimal uptake of nutrients and water while minimizing leaching from the root zone. In addition, drip irrigation can be easily adjusted to suit different types of crops, soil compositions, and topographical conditions. Additionally, this form of irrigation is highly recommended for

sandy soils, which are characterized by a notably reduced water retention capacity. The TDS of water used to regularly irrigate the crop was measured at around 500–700 ppm. The amount of irrigation water was determined based on the FAO-56 method [27]. This method calculates crop water requirement (ETc) by multiplying reference evapotranspiration (ET$_0$) with the crop coefficient (Kc). ET$_0$ is calculated according to the Food and Agriculture Organization of the United Nations (FAO) Penman–Monteith equation, while the seasonal Kc for specific crops can be obtained from the FAO database.

### 2.2. Forward Hydrological Modeling

The vertical model of RWU was simulated using the HYDRUS 1D [13] package that solves the Richard equation using an axisymmetric finite-element code for saturated and unsaturated water flow. The water flow equation involves a sink term representing RWU. The Richard equation under the atmospheric upper boundary condition at the surface and free drainage lower boundary condition can be expressed as below (Equation (1)):

$$\frac{\partial \theta}{\partial t} = \frac{\partial}{\partial z}\left[K\left(\frac{\partial h}{\partial z} + 1\right)\right] - S,  \tag{1}$$

where $\theta$ is the volumetric water content (cm$^3$ cm$^{-3}$), h is the water pressure head (cm), t is time (day), z is the spatial coordinate (cm), K is the unsaturated hydraulic conductivity (cm day$^{-1}$), and S is the sink term in the flow equation (cm$^3$ cm$^{-3}$ day$^{-1}$).

The van Genuchten–Mualem constitutive relationships [13] were employed to determine the soil water retention curve as shown below (Equation (2)):

$$\theta(h) = \begin{cases} \theta_r \frac{\theta_s - \theta_r}{[1 + |\alpha h|^n]^m} & h < 0 \\ \theta_s & h \geq 0 \end{cases}$$

$$K(h) = K_s S_e^l \left[1 - \left(1 - S_e^{1/m}\right)^m\right]^2,  \tag{2}$$

$$S_e = \frac{\theta - \theta_r}{\theta_s - \theta_r}$$

where $\theta_r$ is the residual water content (cm$^3$ cm$^{-3}$), $\theta_s$ is the saturated water content (cm$^3$ cm$^{-3}$), $\alpha$, n, and $l$ are empirical parameters, $S_e$ is the effective saturation, and $K_s$ is the saturated hydraulic conductivity (cm/d).

The root water uptake process in HYDRUS 1D is simulated using the Feddes model [28] without considering the osmotic stress that can be expressed as follows (Equation (3)):

$$S(z, t) = \alpha(h, z)\beta(z)T_p$$

$$\alpha(h, z) = \begin{cases} 1 & h_1 \leq h \leq 0 \\ \frac{h - h_2}{h_2 - h_1} & h_2 \leq h \leq h_1 \\ 0 & h \leq h_2 \end{cases}  \tag{3}$$

where $\alpha(h,z)$ is the water stress response function, $\beta(z)$ is the standardized RWU distribution function, and $T_p$ is defined as the potential crop transpiration rate, while $h_1$ and $h_2$ are the soil water potential at which RWU rate decreases from 1 cm and decreases to 0 cm, respectively.

In this study, the parameters of the lemon RWU configuration provided by the HYDRUS 1D were selected. The model was also calibrated through soil hydraulic parameters [13].

### Boundary Conditions

The upper boundary conditions were assigned as atmosphere boundary conditions with surface runoff since the soil profile was subjected to rainfall, evaporation, runoff and root water uptake, and deep drainage [29], while the lower boundary condition was

defined as free drainage since this measurement was conducted into a permeable bottom side experimental box. Based on the above, the water table was far below the root zone which represents the real condition in Saudi Arabia. Results from geophysical acquisition were utilized to determine the hydrological initial conditions.

The meteorological boundary parameters at the hourly time scale (air temperature, wind speed, and net surface radiation) at a specific period (23 March, 11 April, 14 May, 1 June, 11 July, and 21 August) were retrieved from ERA5-Land following [30]. Another meteorological parameter, relative humidity, was estimated based on the air temperature and dew point derived from the same dataset. Those parameters were used to calculate potential evapotranspiration using the Penman–Monteith method at site location. The units of net surface radiation and air temperature were converted from $J/m^2$ to $MJ/m^2$ and K to $^\circ$C, respectively. ERA5-Land is a reanalysis dataset developed by the European Centre for Medium-Range Weather Forecasts (ECMWF) that combines observation data across the world and provides reliable estimates of historical climate [31].

### 2.3. Soil Moisture Profile (SMP) Calculation

Precise estimation and monitoring of the soil moisture profile (SMP) in the root zone is challenging since SMP is contingent upon numerous external factors, predominantly weather conditions and climate fluctuations. Therefore, it is crucial to comprehend the most efficient techniques for evaluating soil moisture levels. The measurements can be accomplished by conducting soil laboratory measurements, using in situ sensors integrated with local meteorological data, and applying advanced satellite technology.

The soil laboratory measurement is known as gravimetric soil moisture detection. This technique utilizes evaporation, flushing, and a chemical reaction to remove water from a soil sample. The gravimetric soil moisture is determined by quantifying the disparity in weight between the wet and dry soil samples. It can be used once as a reference measurement for a study area, but the method is time-consuming and cannot be used for monitoring purposes.

For soil moisture measurements and monitoring with sensors, tensiometers, gypsum blocks, and time-domain (TDR) or frequency-domain (FDR) reflectometer probes can be used to determine and monitor the soil moisture (1D information) at the installation's location. Tensiometers have a significant drawback in that they require frequent maintenance. The gypsum block is better suited for a broader range of soil moisture levels compared to the tensiometer. However, it has a more delicate structure, necessitating regular replacement. The reflectometers rely on geoelectrical measurements, making them a more resilient method that does not necessitate frequent maintenance. Nevertheless, the process of analyzing data using them is more intricate. Furthermore, specific calibration is required to align with ground characteristics.

In addition, the advanced satellite technology incorporating the existing passive microwave satellite missions, Soil Moisture and Ocean Salinity (SMOS), and Soil Moisture Active Passive (SMAP), can solely estimate soil moisture at different scales (local, regional, global) in the topmost 5 cm layer. The combined use of different bands in satellite images showed the potential to determine the global soil moisture profile precisely up to a depth of 30 cm [32].

Agricultural practitioners can utilize the aforementioned technologies to enhance irrigation efficiency, resulting in reduced water consumption and decreased production expenses. In order to achieve optimal accuracy for different soil types, it is essential to perform a calibration that is specific to the soil. This is because the sensor's measurements are influenced not only by the water content of the soil, but also by its physical and chemical properties. In our experimental setup, the soil type is sandy. The sandy soil texture allows rapid infiltration and percolation of precipitation, thereby reducing the rapid loss of soil moisture by the uptake and transpiration of water by plants. Implementing these strategies for sandy soils is exceedingly challenging due to high drainage rates and difficulties in achieving a uniform water distribution throughout the soil column.

To avoid the installation of TDR/FDR sensors to estimate the soil moisture locally (where the sensors were installed), a more comprehensive geoelectrical resistivity tomography (ERT) experiment was set up and tested. The ERT application data offer insights into the subsurface's resistivity distribution, enabling analysis of its composition and structure. Additionally, ERT can be utilized to monitor dynamic processes, such as fluid flow, over time, which is called time-lapse geoelectrical monitoring. For the ERT data collection, a minimum of four electrodes is required: two to inject current into the ground, and two to measure the resultant potential difference. The soil's 2D and/or 3D resistivity models are obtained by combining measurements from multiple electrode pairs with varying spacing along the experimental geometry. The geoelectrical models are influenced by factors such as, lithology, porosity, structure, temperature, root density, and water content [33–36].

Time-Lapse 3D ERT Acquisition

ERT surveys typically involve a number of electrodes to estimate the resistivity distribution in the subsurface. A pair of electrodes is used to inject current into soil, while another pair is used to measure the potential difference; thus, the apparent resistivity can be estimated. The real resistivity distribution in 2D, 3D, or 4D can be constructed through inverse modeling based on multiple combinations of current and potential electrodes.

To collect detailed soil moisture data for this research, monthly time-lapse 3D geoelectrical data were collected within 7 months (March–September 2023) (Table 1) with the highest accuracy in a wooden tank to extract the average soil moisture profile. This profile will be used to estimate the RWU using the HYDRUS modeling code. We installed an electrical resistivity setup of 8 boreholes and 4 pairs of crosshole measurements, each containing 24 electrodes at 5 cm intervals, totaling 192 electrodes. It should be mentioned that the boreholes were installed at about 50 cm from the sides and the bottom of the tank to avoid any side effects during the data acquisition and processing. These were strategically placed for comprehensive 3D coverage, as illustrated in Figure 2. Preserving electrode positions was crucial for accurate geometric factor estimation and data interpretation. Data acquisition employed the Syscal-Pro system, and Res3dinvx64 version 3.20.0 software processed the raw data to produce 3D sections. The selected protocols were instrumental in achieving a higher resolution of the collected data while expediting the measurement process, which is crucial for effectively capturing dynamic processes in the root zone. A dipole–dipole diagonal electrode array was selected for the borehole pairs A1–A2, A3–A4, B1–B2, and B3–B4 and merged in a single file for processing the data in 3D mode. In processing, we refined data for accuracy, removing outliers and precisely defining tank boundaries. We used proper regularization parameters, such as a damping factor and robust data constraint, including the L1-norm with adjustments, to mitigate side cell effects and maintain resistivity accuracy near boreholes. A total of 14 datasets were acquired during the study period, coming from the boreholes A and B. In total, 9270 readings were collected for each borehole group (A1–A4 or B1–B4), which required 7.5 h to complete the measurement. Thus, each 3D resistivity model consists of 10,540 measurements prior to filtering. After the filtering, less than 16% of the data (noisy or low-quality data) were removed, and the rest was used for further processing.

**Table 1.** Dates of 3D ERT monitoring at site location.

| Date | Acquisition Number |
|---|---|
| 22 February 2023 | 0 (background) |
| 23 March 2023 | 1 |
| 11 April 2023 | 2 |
| 14 May 2023 | 3 |
| 1 June 2023 | 4 |
| 11 July 2023 | 5 |
| 21 August 2023 | 6 |
| 19 September 2023 | 7 |

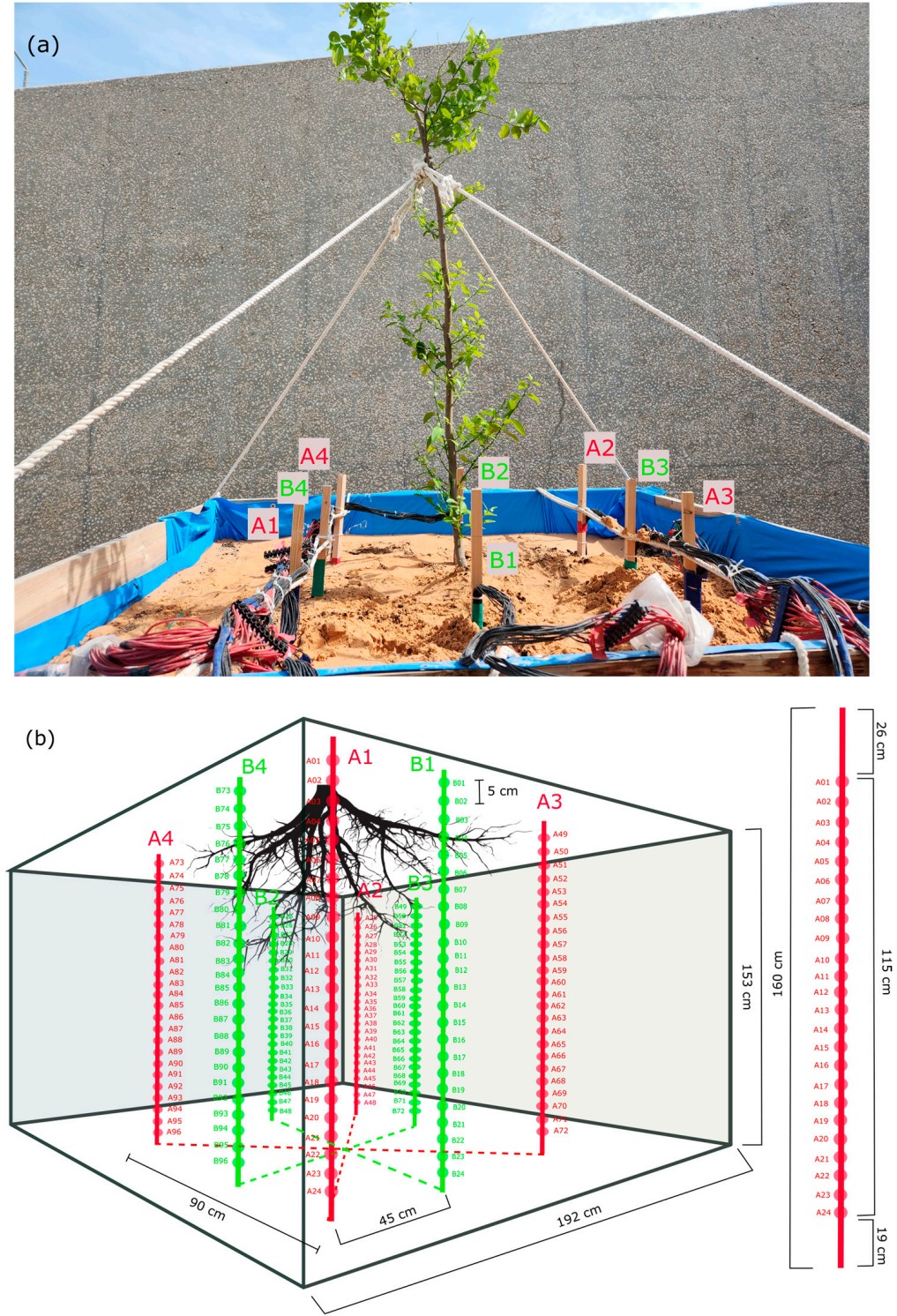

**Figure 2.** (**a**) A photograph of the controlled experiment depicting the lemon tree positioned at the center of the box, surrounded by eight boreholes containing electrodes and (**b**) a schematic of the experimental 3D resistivity geometry illustrating the boreholes groups A1–A4 (red) and B1–B4 (green) and electrodes' configuration in the subsurface.

## 3. Results and Discussion

After irrigation, the soil moisture profile (SMP) needs to be defined as the initial condition. The SMP is indirectly calculated using the non-destructive ERT method. The way that ERT was used to calculate a detailed and accurate SMP is described in the following section. The simulation of RWU was conducted at an hourly time step. In addition, to reflect the diurnal variation, the simulation was designed at 08:00, 11:00, 14:00, 17:00, and 22:00.

### 3.1. Soil Moisture Profiles Derived from ERT

Figure 3 displays the final 3D ERT processed models after seven iterations and a final root mean square (RMS) of 10.2%. This 3D resistivity model is one of the total seven time-lapse tomographic models performed during this research. The colors represent the spatial resistivity (soil moisture) distribution in five depth slices: 3–5 cm, 9–12 cm, 17–22 cm, 28–34 cm, and 42–51 cm. Hot (red) colors indicate high-resistivity areas and cold (blue) colors identify low-resistivity anomalies. These depth images show the volume of the root system/zone, as depicted by the dashed circle, as a high-resistivity anomaly due to the complexity of the root zone compared to the homogeneous (low-resistivity) host soil. The most evident (least RMS) tomographic detection of the root zone is shown at the fourth depth slice (28–34 cm), which is in agreement with the expected depth where the colonized root zone is expected to be found [20,26].

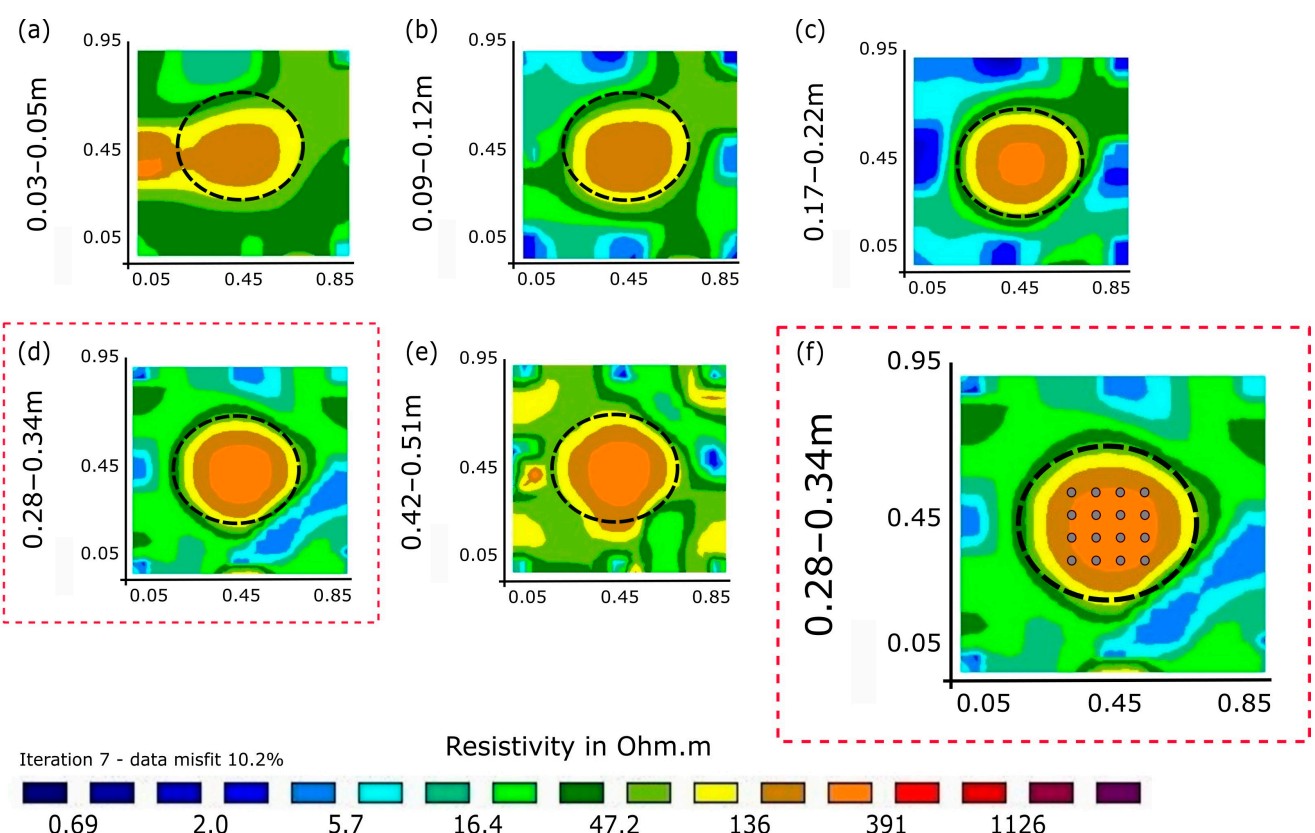

**Figure 3.** Depth slices at of electrical resistivity tomography after seven iterations at different depths: (**a**) 3–5 cm, (**b**) 9–12 cm, (**c**) 17–22 cm, (**d**) 28–34 cm, and (**e**) 42–51 cm. (**f**) The depth slice where the most evident tomographic detection of the root zone (least RMS) was observed. Color indicates spatial resistivity distribution (soil moisture) within soil.

The 3D resistivity model for each of the seven time-lapse periods (from March to August 2023) has a spatial resolution of 5 cm (corresponding to the electrode spacing used) and a vertical resolution of 3–7 cm, depending on the depth. To obtain a representative and accurate soil moisture profile based on multiple resistivity measurements, a grid of 16 nodes (represented by grey dots in Figure 3f) was used to sample the root zone. The average resistivity from these 16 sampling points for 15 depths was extracted from the high-resolution 3D resistivity model, ranging from the surface to a maximum depth of 110 cm, and provides a comprehensive characterization of the soil moisture dynamics (Figure 4). Thus, 240 resistivity points (15 resistivity measurements for each of the 16 soundings used) were used to determine the soil moisture instead of using one or two TDR sensors that, probably due to the soil type, will not be enough to define the SMP.

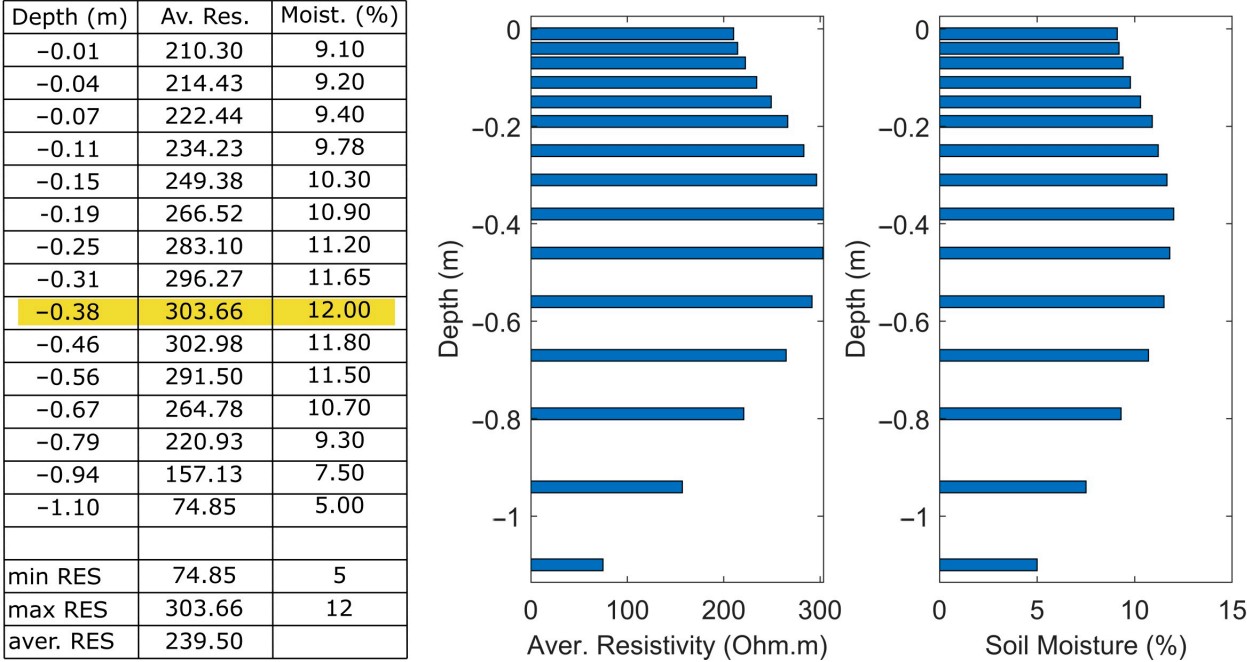

| Depth (m) | Av. Res. | Moist. (%) |
|---|---|---|
| −0.01 | 210.30 | 9.10 |
| −0.04 | 214.43 | 9.20 |
| −0.07 | 222.44 | 9.40 |
| −0.11 | 234.23 | 9.78 |
| −0.15 | 249.38 | 10.30 |
| −0.19 | 266.52 | 10.90 |
| −0.25 | 283.10 | 11.20 |
| −0.31 | 296.27 | 11.65 |
| −0.38 | 303.66 | 12.00 |
| −0.46 | 302.98 | 11.80 |
| −0.56 | 291.50 | 11.50 |
| −0.67 | 264.78 | 10.70 |
| −0.79 | 220.93 | 9.30 |
| −0.94 | 157.13 | 7.50 |
| −1.10 | 74.85 | 5.00 |
| | | |
| min RES | 74.85 | 5 |
| max RES | 303.66 | 12 |
| aver. RES | 239.50 | |

**Figure 4.** The average resistivity and soil moisture profile from all measuring periods derived from ERT. The yellow-shaded cells also depict the highest resistivity and soil moisture, showing the center of the root zone.

Based on the vertical distribution of the resistivity during the experimental period, the maximum resistivity is depicted at a depth of 38 cm (yellow-shaded cells in Figure 4), which agrees with the literature [20,26] and the minimum resistivity (74.85 Ohm.m) at the bottom of the model, where the water is accumulated. The parabolic shape of the resistivity distribution with depth (graph in the middle in Figure 4) is consistent with the study by Mishra et al. [37] since the measurements were always acquired in a short time after the irrigation to ensure good contact resistance and the collection of high-quality resistivity data.

Since soil moisture profiles are required as input data for estimating RWU using HYDRUS 1D, attempts were made to convert the final vertical resistivity distribution into water content or soil moisture. The average resistivity profile (Figure 4) ranges from 74.85 to 303.66 Ohm.m. Given the expected soil moisture content in sandy soils ranging from 5% to 12%, the soil moisture profile derived from the resistivity distribution was estimated and graphically presented (last graph on the right in Figure 4).

### 3.2. Model Calibration

Considering the results of soil mechanical composition, soil hydraulic characteristics were estimated through the ROSETTA neural network prediction model and hydraulic conductivity test. In turn, the estimated values of soil hydraulic properties were calibrated using the time-varying soil moisture profile values provided by the aforementioned ERT geophysical technique. The soil hydraulic parameters were as follows: Van Genuchten parameters ($\theta_r$, $\theta_s$, $\alpha$, n, Ks, l), as listed in Table 2. During the calibration process, a good agreement between the soil moisture profile values provided by the ERT method and the HYDRUS 1D simulation results was demonstrated based on the root mean square error (RMSE) and correlation coefficient (R) statistical indices, indicating a successful calibration process (Figure 5).

**Table 2.** Soil hydraulic parameters.

| Depth (cm) | $\theta_r$ | $\theta_s$ | $\alpha$ | n | $K_s$ (cm/d) | l |
|---|---|---|---|---|---|---|
| 0–100 | 0.045 | 0.3771 | 0.0348 | 4.3305 | 18.87 | 0.5 |

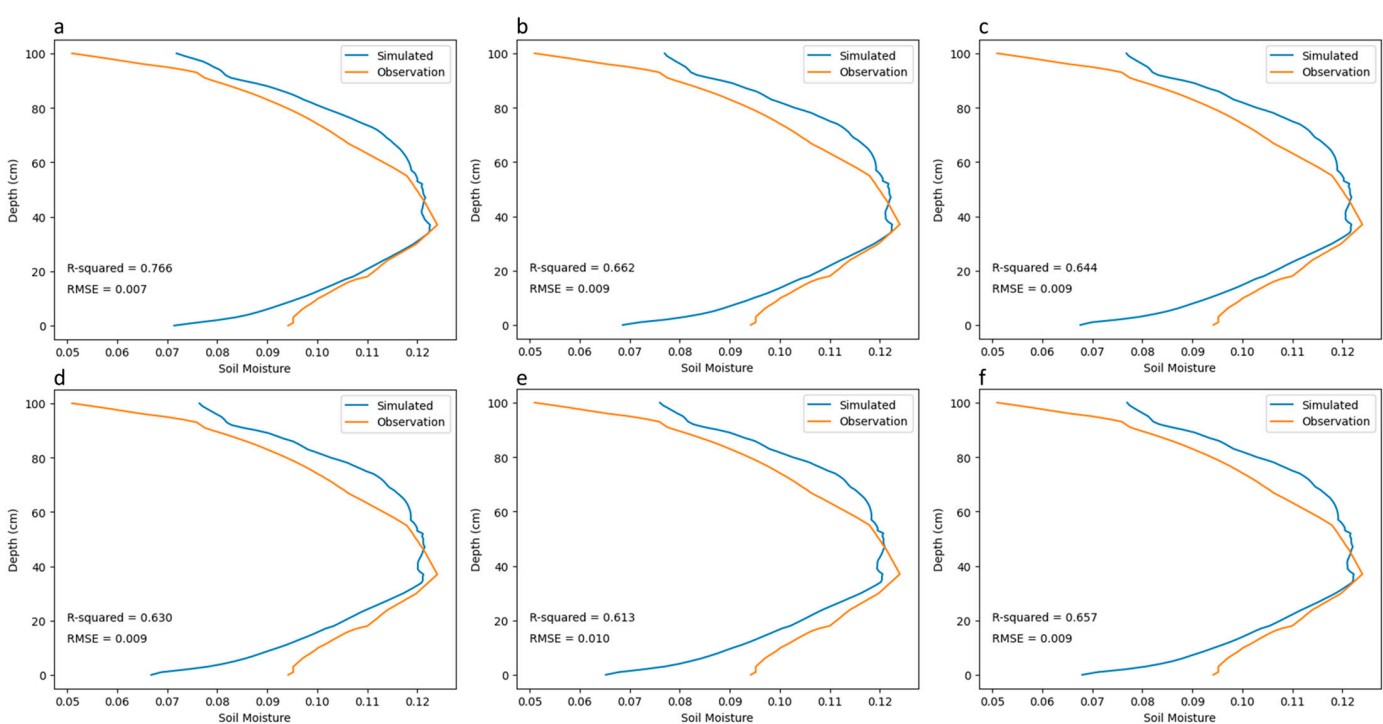

**Figure 5.** HYDRUS 1D calibration results for each month: (**a**) March, (**b**) April, (**c**) May, (**d**) June, (**e**) July, and (**f**) August. Good agreements between calibrated model with observed soil moisture were observed.

### 3.3. Simulated Soil Water Content

The calibrated model was employed to simulate soil water content at a depth of 0–100 cm on an hourly time step for each month (Figure 6). As evident from the simulated soil water content, overall, the soil water content profile exhibited similar patterns with little change over different periods. There is a general trend of decreasing total soil water content over time due to soil water consumption and redistribution. A notable pattern emerges during the initial hours following irrigation when the highest soil water content for each period is consistently observed at a depth of 80–90 cm, before rapidly decreasing, showing a parabolic pattern. Over the following hours, the soil water content profile for each period changed its typical vertical pattern, with water content gradually increasing with depth, and the highest soil water content was predominantly located at depths of

80–100 cm. Overall, the moisture content at a depth of 0–30 was significantly lower than the soil profile during the different periods.

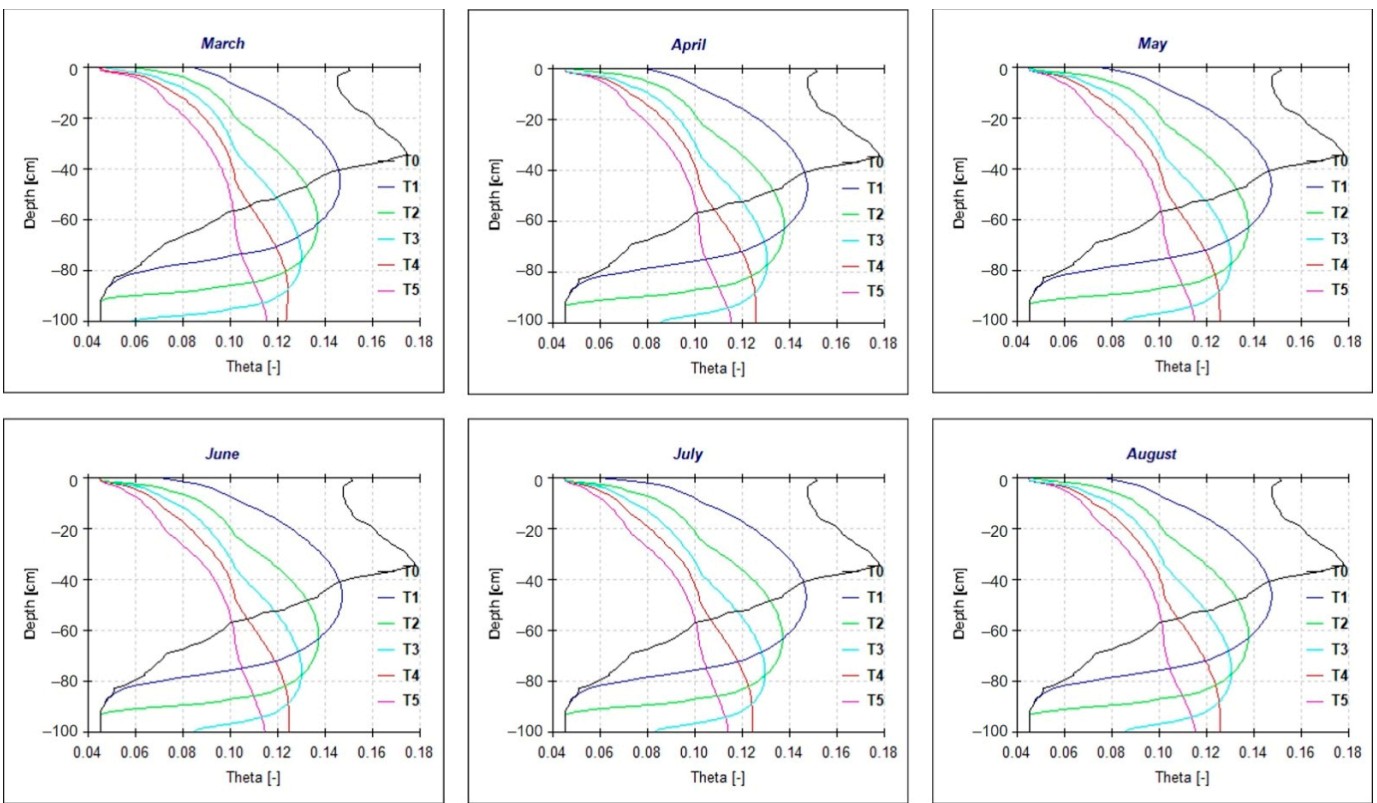

**Figure 6.** Simulated soil water content on 23 March, 11 April, 14 May, 1 June, 11 July, and 21 August 2013. T0 is the initial condition, while T1, T2, T3, T4, and T5 represent the time intervals at 08:00, 11:00, 14:00, 17:00, and 22:00, respectively.

*3.4. Diurnal and Seasonal Variations in RWU*

The diurnal dynamics of RWU simulated by HYDRUS 1D from March to August 2023 are illustrated in Figure 7a. The simulated RWU for each month exhibited fluctuations with typical peaks during the morning period from 08:00 to 11:00. Specifically, RWU increased during the first several hours of sunlight, reaching a maximum before midday, and then decreased until the end of the observation period (22:00). This diurnal variation might indicate that the morning period can be considered as the best irrigation timing when RWU is the highest. At the seasonal time scale, the hourly RWU values were generally larger in July, resulting in larger cumulative RWU values, as shown in Figure 7b, while the most negligible RWU variation was observed in March. RWU values started to increase in March, reaching their peaks in July before and decreasing again until August.

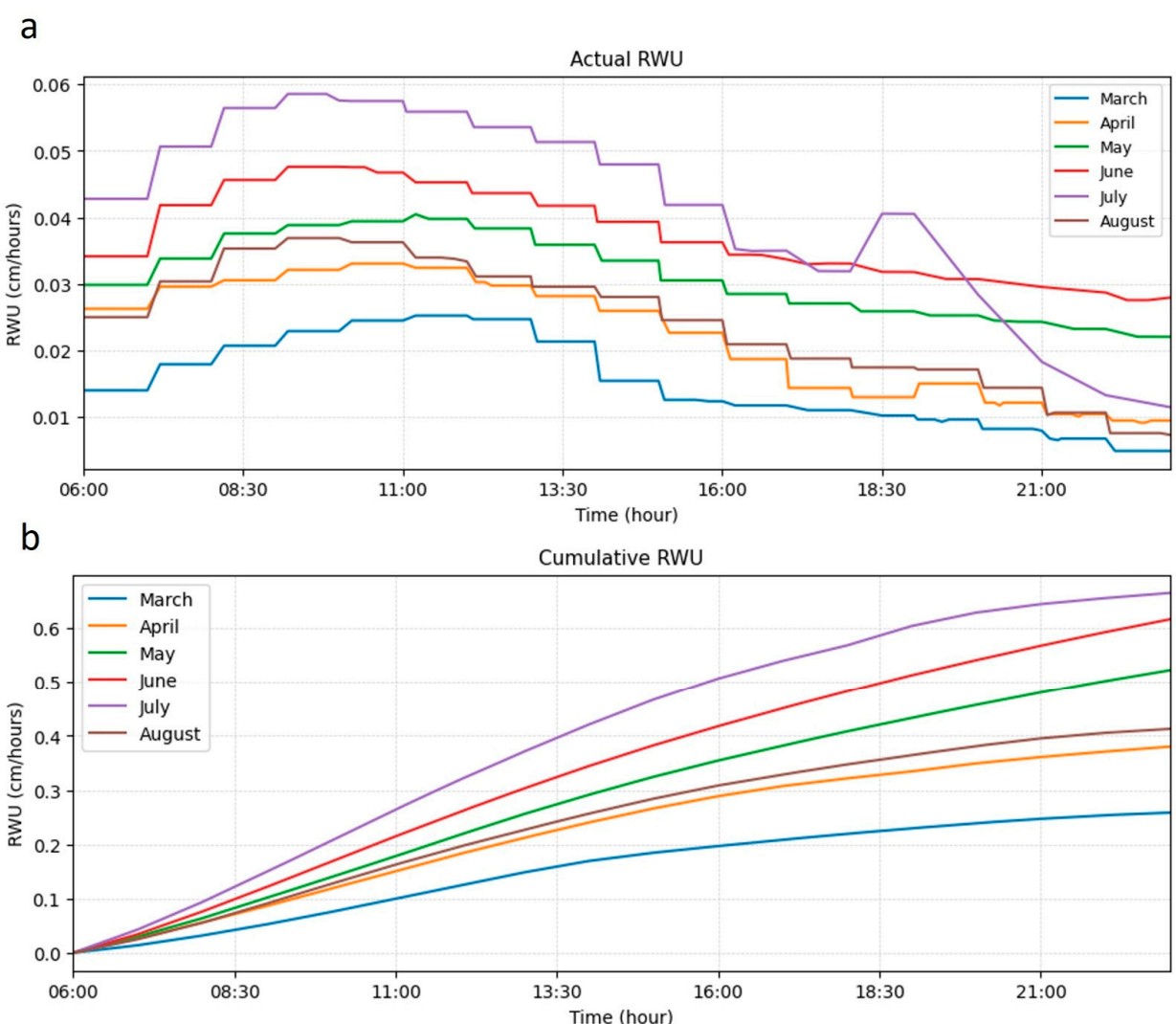

**Figure 7.** (**a**) The changes in RWU rates and (**b**) cumulative RWU in different months. The simulation period spans from 06:00 in the morning to 22:00 at night.

### 3.5. RWU Rate at Different Soil Depths

The RWU rate based on the Feddes model is depicted across different depths and times, providing insights into the temporal and spatial patterns of water extraction by plant roots (Figure 8). Overall, RWU primarily occurred in the upper 30 cm layer of the soil profile, reflecting the actual concentration of roots in this area. This also explains why the lowest soil water content was always observed at the upper soil profile (depth of 0–30 cm) compared to the rest of the soil layer (depth of 30–100 cm). Following irrigation, the RWU zone remained within the depth of 0–30 cm during the initial hours. However, as time and temperature increased, the RWU zone gradually shifted from the upper to the lower soil profile. The most significant shift of the RWU zone occurred during the summer periods. In June and July, higher RWU rates were predominantly distributed at depths of 4–30 cm and 7–30 cm beneath the soil surface, respectively, suggesting an increase in soil evaporation during mid-day. Therefore, managing irrigation water during the summer months in accordance with RWU zone is essential to avoid water loss through evaporation.

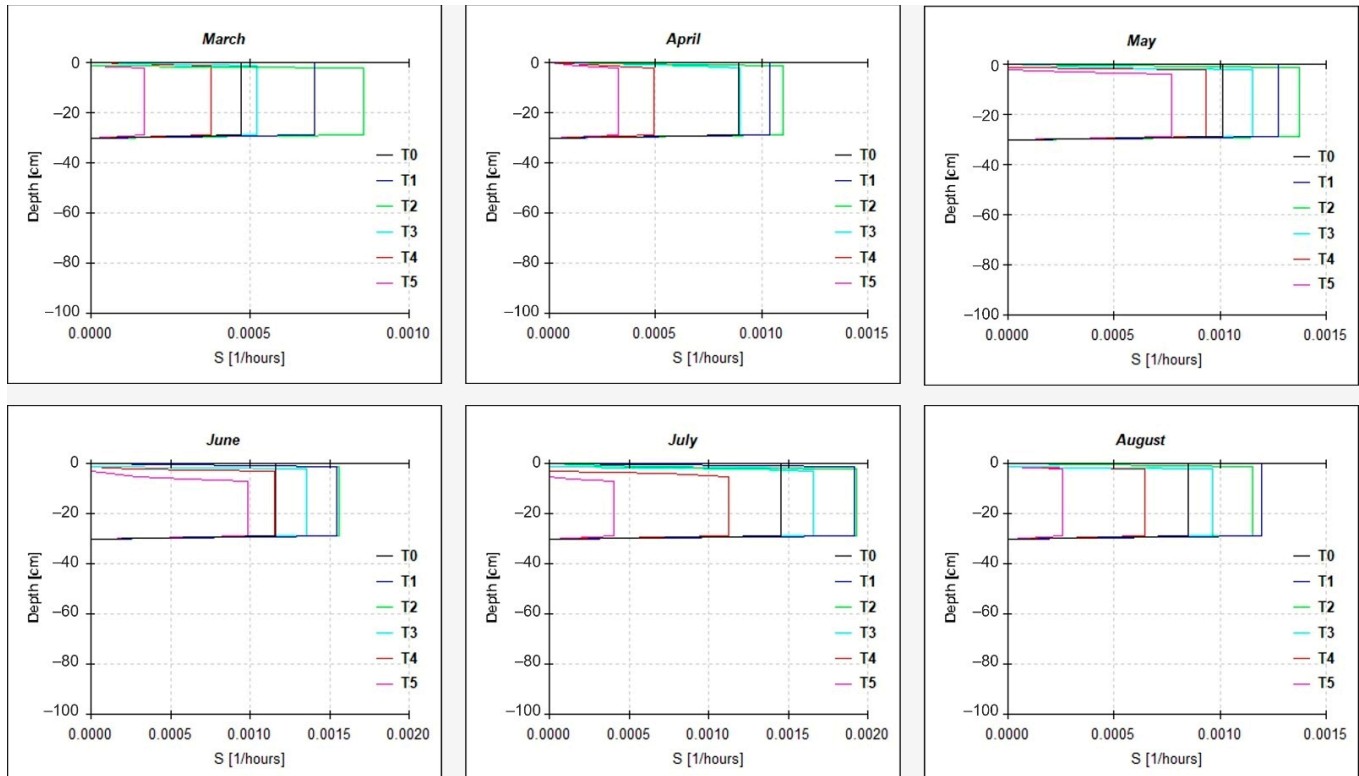

**Figure 8.** RWU rates at various soil depths and time periods. The designation 'T0' represents the initial condition, while 'T1' through 'T5' correspond to time intervals at 08:00, 11:00, 14:00, 17:00, and 22:00, respectively.

## 4. Conclusions

Effective management of citrus irrigation and nutrient allocation is crucial, particularly in regions facing limited water resources, such as Saudi Arabia. Limited amounts of irrigation water or low-quality water are factors that have a detrimental impact on the productivity of citrus trees and the quality of their fruit. When it comes to fertilizing citrus plants, nitrogen is the most important nutrient. The elevated utilization of nitrogen fertilizers in citrus cultivation leads to the leaching of nitrates from the area surrounding the roots, which poses a risk of contaminating groundwater. Hence, effectively managing irrigation and fertilization schedules in citrus farming poses significant difficulties in minimizing water and nitrogen losses beneath the root zone. Consequently, the use of efficient simulation models like HYDRUS has become highly valuable for analyzing the movement of water and solutes in the soil, with the aim of designing effective irrigation and fertigation systems for citrus farming. To use HYDRUS for modeling RWU, the soil moisture content is needed. Both destructive (gravimetric) and nondestructive techniques (soil moisture sensors) can be used to measure the soil moisture content but in sandy soils, both techniques are very difficult. Thus, a new approach/framework for sandy soils is suggested, tested, and evaluated.

In this study, two well-known undisturbed techniques, a numerical one, HYDRUS 1D, and a geophysical one, ERT, were employed to assess the daily and seasonal variations in RWU under a controlled experiment. In this context, multiple time-lapse ERT measurements were utilized to transform the soil electrical resistivity into soil water content. The transformed electrical resistivity values served as input for the initial condition of HYDRUS 1D and model validation. The simulated RWU revealed that the morning period (08:00–11:00) could be considered the optimal irrigation timing, as it corresponded to the highest RWU rates. During summer, RWU zones shifted to lower depths due to increased soil evaporation. The findings of this study have the potential to assist farm stakeholders in

efficiently managing limited freshwater resources, particularly in the face of rising temperatures. Additionally, this study successfully highlighted the critical importance of accurately setting initial conditions for model simulations to obtain reliable results.

Hence, it is imperative to conduct additional research to evaluate a broader range of irrigation schemes in practical settings combining numerical models with geophysical methods. This research can be used as initiative for joint, numerical and geophysical methods, applications, examining the effect of nitrates, the distribution of salinity in soil under various tree crops, soil types, and climatic conditions on plant growth. Conducting such studies would contribute to the enhancement of irrigation and fertigation designs and strategies for perennial crops that are irrigated using drip irrigation systems. This, in turn, would result in more effective and environmentally friendly cropping practices.

**Author Contributions:** Conceptualization, N.N.K. and P.S.; methodology, N.N.K. and P.S.; software, A.P. and P.K.; validation, N.N.K. and P.S.; formal analysis, A.P.; investigation, A.P., Y.M.H.M. and P.K.; resources, P.S.; data curation, A.P., N.N.K., P.K. and P.S.; writing—original draft preparation, A.P., P.K. and P.S.; writing—review and editing, A.P., N.N.K., Y.M.H.M., P.K. and P.S.; visualization, A.P. and P.K.; supervision, P.S.; project administration, P.S.; funding acquisition, P.S. All authors have read and agreed to the published version of the manuscript.

**Funding:** This research received no external funding.

**Institutional Review Board Statement:** Not applicable.

**Informed Consent Statement:** Not applicable.

**Data Availability Statement:** The data presented in this study are available on request from the corresponding author.

**Acknowledgments:** The authors gratefully acknowledge the College of Petroleum Engineering & Geosciences for technical and financial support.

**Conflicts of Interest:** The authors declare no conflicts of interest. The funders had no role in the design of the study; in the collection, analyses, or interpretation of data; in the writing of the manuscript; or in the decision to publish the results.

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
