# Peer review of "Simulating Tree Root Water Uptake in the Frame of Sustainable Agriculture for Extreme Hyper-Arid Environments Using Modeling and Geophysical Techniques"

_sustainability, doi:10.3390/su16083130_

Round 1

Reviewer 1 Report

Comments and Suggestions for Authors

This study used Electrical Resistivity Tomography (ERT) derived soil moisture data to drive the HTDRUS-1D model and understand RWU under controlled experimental conditions in an extreme hyperarid environment. They concluded that the highest RWU rates occurred during the morning (08:00-11:00). RWU activity predominantly concentrated in the upper soil profile (0-30 cm). With increasing temperature, there was a tendency for the RWU zone to shift to lower depths within the soil profile.

In general, this paper is well-written and easy to follow. Therefore, I recommend a minor revision. In addition, I suggest the authors focus on the following to make the conclusions more robust.

Major comments:

1         In Introduction, please clarify the novelty of this study, since we know hydrus has been widely used to understand RWU in different environments. If the novelty is about ERT, the authors should provide some review of this technique and what is new in this study.

2         If drip irrigation was applied, why did the authors use HYDRUS-1D instead of 2D?

3         Please clarify how the resistivity in ERT was converted to soil moisture.

4         Section 3.2: The authors should provide the fit between simulated and observed soil water contents. Otherwise, the readers have no idea about the applicability of ERT or the accuracy of the HYDRUS model in this case study.

Minor comments:

1         All figure captions should be standalone, i.e., descriptive enough to be understood without having to refer to the main text.

2         Figure 2b is blurry.

3         Table 2: parameter l is specified, not calibrated, right?

4         Figure 6: the x label should be in 6:00-22:00 time format instead of numbers.

Comments on the Quality of English Language

fine

Author Response

Reviewer 1_____________________________________________________________________

This study used Electrical Resistivity Tomography (ERT) derived soil moisture data to drive the HTDRUS-1D model and understand RWU under controlled experimental conditions in an extreme hyperarid environment. They concluded that the highest RWU rates occurred during the morning (08:00-11:00). RWU activity predominantly concentrated in the upper soil profile (0-30 cm). With increasing temperature, there was a tendency for the RWU zone to shift to lower depths within the soil profile.

In general, this paper is well-written and easy to follow. Therefore, I recommend a minor revision. In addition, I suggest the authors focus on the following to make the conclusions more robust.

Major comments:

  1. In Introduction, please clarify the novelty of this study, since we know hydrus has been widely used to understand RWU in different environments. If the novelty is about ERT, the authors should provide some review of this technique and what is new in this study.

Authors: Thank you for the comment. The novelty of this study is described in the introduction section, and a few more lines were added to better explain the novelty. The benefits of using ERT for SMP are also described in the last sentence in the paragraph below Figure 3.

  1. If drip irrigation was applied, why did the authors use HYDRUS-1D instead of 2D?

Authors: Dear Reviewer thank you for your comment, as mentioned in the text, we use HYDRUS-1D to simulate the drip irrigation process in agreement with many other related publications that provide satisfactory results, see for example:

https://www.sciencedirect.com/science/article/abs/pii/S0378377416305133 https://www.sciencedirect.com/science/article/abs/pii/S037837742032093X                                             https://www.mdpi.com/2073-4441/11/8/1657 

Also, another very important reason we used the HYDRUS-1D is that HYDRUS-1D unlike 2D is open access software (see new text in lines 112 and113)

  1. Please clarify how the resistivity in ERT was converted to soil moisture.

Authors: The way the ERT was used to soil moisture is described in the last 2 paragraphs above Figure 4. Bur briefly, the idea is that from the high-resolution ERT, the resistivity profile was produced and the knowing the min/max value of the observed resistivities and the min/max values of the expected soil moisture in such soil types, we converted the resistivity distribution to soil moisture change with depth.

  1. Section 3.2: The authors should provide the fit between simulated and observed soil water contents. Otherwise, the readers have no idea about the applicability of ERT or the accuracy of the HYDRUS model in this case study.

Authors: In accordance with reviewer’s comment, we have added additional graph (Figure 5) showing simulated and observed soil moisture.

Minor comments:

  1. All figure captionsshould be standalone, i.e., descriptive enough to be understood without having to refer to the main text.

Authors: In accordance with reviewer’s suggestions, we already revised some of figure captions within the manuscript that are not descriptive enough.

  1. Figure 2b is blurry.

Authors: Figure 2 is revised.

  1. Table 2: parameter l is specified, not calibrated, right?

Authors: Yes

  1. Figure 6: the x label should be in 6:00-22:00 time format instead of numbers.

Authors: Based on the reviewer’s comment, the figure was revised.

Reviewer 2 Report

Comments and Suggestions for Authors

In this study, HYDRUS 1D and ERT were used to simulate RWU under controlled experimental conditions and extreme ultra-dry environment, which is innovative. There are some minor problems that need to be considered and modified by the author:

1. The measurement methods of soil moisture, root system and meteorological data need to be refined. The advantages and disadvantages of different measurement methods for the same index have been compared many times in this paper, but there is too little description of the specific measurement.

2. How to calibrate TDR with ERT, please give the specific implementation method and data interpretation.

3. Abnormal data is removed on line 295. Please specify which data is abnormal and what proportion is abnormal data?

4. Line 335, why choose 38cm? Please explain the reason.

5. On the section of 3.2, the model calibration should be carried out at each layers, and it is meaningless to only correct the average value of 0-100cm.

6. The measured values should be added in Figures 5 and 6, and the accuracy of the model can be verified by comparing the measured values with the simulated values. Supplementary sap flow data is needed to verify RWU, and it is not meaningful to analyze only the simulation results.

Comments on the Quality of English Language

Moderate editing of English language required,  some expressions are too verbose in section of Materials and Methods.

Author Response

Reviewer 2_____________________________________________________________________

In this study, HYDRUS 1D and ERT were used to simulate RWU under controlled experimental conditions and extreme ultra-dry environment, which is innovative. There are some minor problems that need to be considered and modified by the author:

  1. The measurement methods of soil moisture, root system and meteorological data need to be refined. The advantages and disadvantages of different measurement methods for the same index have been compared many times in this paper, but there is too little description of the specific measurement.

Authors: Thank you for your suggestion. We have added more descriptions of the meteorological measurements in section 2.2.1. An overview of the available methods for soil moisture estimation with their advantages and disadvantages is given in Section 2.3. In Section 2.3.1, the indirect estimation of the soil moisture by using the 3D resistivity distribution of the root zone is attempted and shown. Section 3.1 also describes in detail, how the resistivity measurements were converted to soil moisture.

  1. How to calibrate TDR with ERT, please give the specific implementation method and data interpretation.

Authors: Calibrating TDR with ERT is only possible if we use a few TDR sensors while we are collecting the ERT data. Unfortunately, we were not able to use any TDR sensor. Thus, we decided to try to estimate the soil moisture profile that was needed for the RWU, by using the acquired and high-resolution resulting 3D resistivity model of the root zone.

  1. Abnormal data is removed on line 295. Please specify which data is abnormal and what proportion is abnormal data?

Authors: A few lines with more information about the number of data acquired and the data that remained after filtering were added, above the Table 1.

  1. Line 335, why choose 38cm? Please explain the reason.

Authors: The acquired all the 3D resistivity data for all the time periods. After the processing, we found out that for all time periods, the area with the best response to the root zone was at a depth of 38cm below the ground. The same is shown from the average resistivity distribution with depth in Figure 4. This depth also agreed with other experiments' results [23-26].

  1. On the section of 3.2, the model calibration should be carried out at each layer, and it is meaningless to only correct the average value of 0-100cm.

Authors: Thank you for your feedback. In our case, the soil parameter remains consistent throughout, with approximately 99.5% sand and 0.5% silt, reflecting the homogeneous nature commonly found in Saudi Arabia. Therefore, we have focused on the average values within the 0-100 cm depth range.

  1. The measured values should be added in Figures 5 and 6, and the accuracy of the model can be verified by comparing the measured values with the simulated values. Supplementary sap flow data is needed to verify RWU, and it is not meaningful to analyze only the simulation results.

Authors: In accordance with reviewer’s comment, we have added an additional graph (Figure 5) showing simulated and observed soil moisture. Since the scope of our study is to promote the use of ERT, which can provide parameters required by Hydrus, we did not utilize sap flow data in our experiment. Again, we do appreciate your feedback.
